# MLLMCLIP: Feature-Level Distillation of MLLM for Robust Vision-Language Representations

## Abstract

Large-scale pretrained vision–language models, such as CLIP, have become the backbone of modern zero-shot recognition. Despite their strong generalization ability, these models often struggle with compositionality, particularly in understanding attribute-object combinations and relational structures. Recent studies mitigate this issue by augmenting training with synthetic hard negatives generated by large language models and text-to-image models. Yet, this strategy relies on separate expert models, introducing a sequential generation pipeline with quality-control overhead and resulting in a disjointed source of multimodal understanding. To overcome these limitations, we propose MLLMCLIP, a feature-level distillation framework that bypasses synthetic data generation by directly transferring multimodal knowledge from Multimodal Large Language Model (MLLM). Our framework addresses the key challenges of cross-architecture distillation with three core contributions: (1) a question-answering-based protocol to select the teacher MLLM, (2) an attention-based method to identify salient teacher tokens, and (3) the successful adaptation of Centered Kernel Alignment for stable knowledge transfer. MLLMCLIP achieves state-of-the-art performance on 9 out of 11 compositionality benchmarks, while also yielding significant improvements in general-purpose tasks, such as zero-shot classification and image-text retrieval.

## 1 Introduction

The emergence of CLIP (Radford et al., 2021) marked a turning point in vision-language learning, showing strong performance in zero-shot classification and retrieval. Following its success, a wide range of approaches have been proposed to enhance CLIP, with a focus on embedding space (Goel et al., 2022; Lavoie et al., 2024), data efficiency (Li et al., 2021; Joshi et al., 2024), and higher-quality captions (Fan et al., 2023; Lai et al., 2024). Despite these advances, a critical limitation has emerged: CLIP often exhibits a bag-of-words behavior (Yuksekgonul et al., 2022), which prevents it from capturing the compositionality of language.

Constructing hard negatives that require compositional reasoning is a common direction to mitigate this issue. Recent studies (Wu et al., 2023; Patel et al., 2024; Singh et al., 2025) implement this approach by leveraging Large Language Models (LLM) and text-to-image models to synthesize hard negative pairs, which are then incorporated into the CLIP training process. However, these approaches still suffer from two key limitations: (1) the lack of unified multimodal understanding, as they rely on separate generative models; and (2) the inherent inefficiency of their sequential data generation pipeline. These shortcomings motivate an alternative direction: directly leveraging the rich, unified understanding of Multimodal Large Language Models (MLLM) as a source of additional knowledge. Specifically, distilling knowledge at the feature level, rather than synthesizing new data, can bypass the inefficient generation pipeline and enable a more direct and efficient transfer of semantics without iterative sampling or quality control.

While the concept of leveraging MLLM to enhance CLIP is straightforward, building a practical distillation framework hinges on several key design considerations. The first step is selecting a suitable teacher model. Our analysis of MLLM on compositional benchmarks reveals that some MLLM exhibit positional bias, and larger parameter counts do not guarantee stronger compositional under-

standing. The second is determining the teacher tokens. Most distillation research has focused on homologous architectures, transferring knowledge from large models to their smaller counterparts in domains such as CLIP (Yang et al., 2024; Chen et al., 2024), LLM (Chenglin et al., 2024; Xu et al., 2024b), and MLLM (Cai et al., 2024; Xu et al., 2024a). In contrast, distilling knowledge from the generative decoder of MLLM into the representational encoders of CLIP involves a fundamental architectural mismatch. This heterogeneity complicates the selection of teacher tokens, as there is no direct correspondence between the layers or feature spaces of the two models. The third consideration is the distillation loss. While existing methods are primarily built upon direct similarity (e.g., MSE) and relational objectives (e.g., Centered Kernel Alignment (CKA)), their effectiveness for distillation between heterogeneous, multimodal models has been largely unexplored.

In this paper, we propose MLLMCLIP, a novel distillation framework designed to effectively transfer the rich multimodal understanding of MLLM into CLIP's encoders. Each challenge above is addressed with our key components: (1) The comprehensive evaluation protocol based on question-answering reformulation to identify MLLM with superior compositional understanding to act as the teacher. (2) The attention-based method to select the most salient features to serve as teacher tokens. (3) The extension of CKA to multimodal, cross-architecture distillation, enabling stable transfer of structural and semantic knowledge.

We evaluate our method on 11 compositionality benchmarks, where it outperforms previous approaches on 9 tasks and achieves the highest average score. Beyond compositional reasoning, our method delivers consistent gains in zero-shot classification and image–text retrieval, demonstrating enhanced generalizability across tasks. Extensive ablation studies further verify the contribution of each component to the overall performance. The contribution of our paper is summarized as follows:

- To the best of our knowledge, we propose the first distillation framework that leverages multimodal interaction signals from MLLM into CLIP's encoders, successfully bridging a fundamental architectural mismatch.
- Our method addresses the core challenges of this task by introducing a novel protocol for teacher selection, an attention-based token selection method, and a robust distillation loss.
- MLLMCLIP achieves state-of-the-art performance on 9 out of 11 compositionality benchmarks, along with notable gains in zero-shot classification and image-text retrieval.

## 2 RELATED WORKS

### 2.1 COMPOSITIONALITY

The success of CLIP has inspired numerous studies (Mu et al., 2022; Lavoie et al., 2024; Zheng et al., 2024) aimed at enhancing its generalizability through data augmentation, improved training strategies, and enriched textual supervision, such as incorporating paraphrased or longer captions during training. In contrast to these general enhancements, Yuksekgonul et al. (2022) identifies a key limitation of CLIP: its tendency to behave like a bag-of-words model. This observation introduces the issue of compositionality, which has led to the development of new benchmarks (Krojer et al., 2022; Peng et al., 2024; Dumpala et al., 2024) specifically designed to assess compositional reasoning in vision-language models. To address this problem, several methods have been proposed that utilize hard negative pairs for contrastive learning. Prior work has explored several strategies for constructing hard negatives. Early works employ WordNet (Fellbaum, 2010) to generate semantically challenging negative texts (Yuksekgonul et al., 2022; Oh et al., 2024). Subsequent methods utilize LLMs to synthesize negative texts and further leverage text-to-image generative models to produce corresponding negative images, enabling sequential construction of multimodal negatives (Wu et al., 2023; Patel et al., 2024; Singh et al., 2025).

### 2.2 MULTIMODAL LARGE LANGUAGE MODEL

Recent advances in large language models have led to the development of MLLM, which can process and reason across multiple modalities. Models such as LLaVA (Liu et al., 2024), LLaMA-Vision (Grattafiori et al., 2024), InternVL (Zhu et al., 2025), and Qwen-VL (Bai et al., 2025) demonstrate strong performance by aligning image features with the token-based input space of LLMs, typically through a visual projection module. While early MLLMs relied on encoder-based

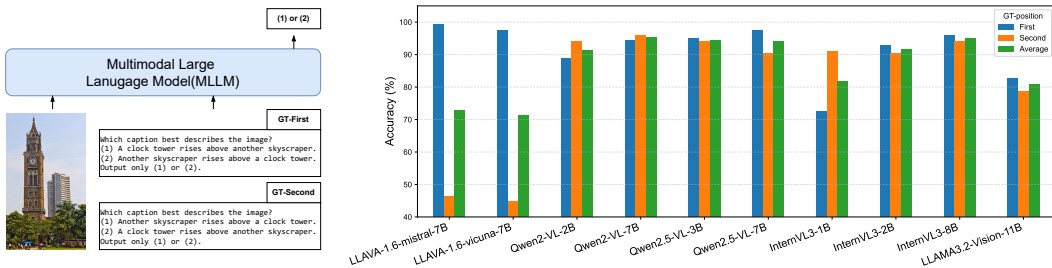

Figure 1: Evaluation of MLLMs on compositional benchmarks. (**Left**) Illustration of the evaluation protocol, where models are asked to choose the correct description when the ground-truth is presented in different orders. (**Right**) Performance of various MLLMs, reported under different ground-truth positions. Detailed numerical results are available in Table A.

visual backbones, recent work has explored encoder-free architectures (Wang et al., 2023; Diao et al., 2024; Luo et al., 2025) that embed image patches directly as input tokens, offering greater flexibility in input format. Our framework is agnostic to these architectural choices, capable of leveraging the multimodal understanding from both encoder-based and encoder-free MLLMs as a teacher.

## 3 METHOD

### 3.1 TEACHER MODEL SELECTION

To select a suitable teacher, we evaluate MLLMs using the SugarCrepe benchmark (Hsieh et al., 2023), which is designed to test compositional understanding in vision-language models. The benchmark provides an image paired with a correct and a perturbed caption, and is typically used by selecting the caption with the higher image-text similarity score. As MLLMs are not assessed via simple similarity scores, we adapt this benchmark by reformulating it as a question-answering (QA) task. For example, the model is prompted with an image and text such as:

*"Which caption best describes the image? (1) The horse is eating the grass. (2) The grass is eating the horse. Output (1) or (2)."*

Our evaluation includes recent open-source MLLMs with parameters ranging from 1B to 11B, such as LLaVA-1.6 (Liu et al., 2024), Qwen2.5-VL (Bai et al., 2025), InternVL3 (Zhu et al., 2025), and LLaMA-3.2-Vision (Grattafiori et al., 2024). To account for position bias, each model is evaluated twice by varying the position of the ground-truth caption. As illustrated in Figure 1, LLaVA-1.6 exhibits strong position sensitivity, showing a large accuracy gap depending on caption order. While InternVL3-8B achieves the highest average accuracy, Qwen2.5-VL-3B exhibits minimal position bias and performs comparably, with significantly lower computational cost. Based on this superior balance of performance and efficiency, we adopt Qwen2.5-VL-3B as the teacher model for all subsequent experiments.

### 3.2 ARCHITECTURE

#### 3.2.1 STUDENT EMBEDDING (CLIP)

The student model consists of separate CLIP-based image and text encoders, denoted as $f^I$ and $f^T$, respectively. We follow the standard CLIP-style input processing: patch embedding followed by prepending a [CLS] token for image input $x^I$, and tokenization with [SOS] and [EOS] for text input $x^T$. Each sequence is passed through its encoder to produce layer-wise hidden states:

$$h_l^I \in \mathbb{R}^{d_s^I}, \quad h_l^T \in \mathbb{R}^{d_s^T}, \quad l = 1, \ldots, L_s, \tag{1}$$

where $h_l^I$ and $h_l^T$ denote the hidden states of the [CLS] and [EOS] tokens from the $l$-th layer of the image and text encoders, respectively. Here, $L_s$ is the number of encoder layers, and $d_s^I, d_s^T$ are the hidden dimensions of the image and text encoders. To align the feature dimensions of the teacher

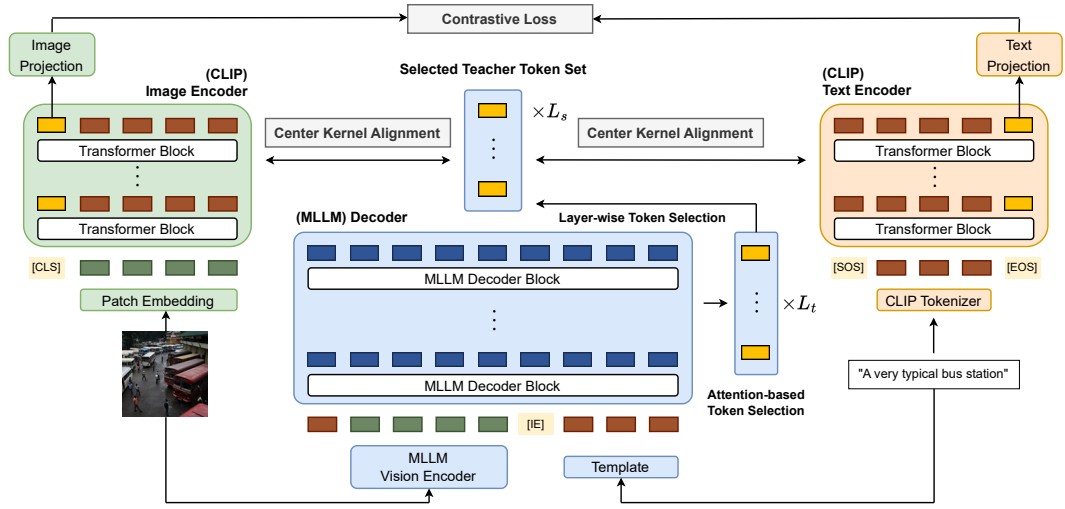

Figure 2: Overview of the MLLMCLIP framework. Student features are passed through auxiliary layers before alignment, and attention-based token selection obtains salient teacher tokens.

and student, the hidden states are passed through auxiliary layers ($\text{Aux}^I$, $\text{Aux}^T$) implemented as a linear layer followed by Layer Normalization:

$$h_l^{\text{Stud},I} = \text{Aux}^I(h_l^I), \quad h_l^{\text{Stud},T} = \text{Aux}^T(h_l^T), \quad h_l^{\text{Stud},I}, h_l^{\text{Stud},T} \in \mathbb{R}^D. \tag{2}$$

These student representations are used to match the corresponding teacher representations. In parallel, the final-layer representations are projected into a common embedding space for contrastive learning. Following the original CLIP architecture, these projection heads, $\text{Proj}^I$ and $\text{Proj}^T$, are implemented as a single linear layer:

$$z^I = \text{Proj}^I(h_{L_s}^I), \quad z^T = \text{Proj}^T(h_{L_s}^T), \tag{3}$$

where $z^I, z^T \in \mathbb{R}^d$.

### 3.2.2 TEACHER EMBEDDING (MLLM)

The teacher model jointly encodes image $x^I$ and $x^T$ as a unified sequence. The image is first tokenized via patch embedding and processed through a vision encoder, which is part of the MLLM architecture. The text input is inserted into a fixed template designed to activate the model's compositionality, as detailed in Appendix B.2. We concatenate the image features with the `[Image End]` token, followed by the text tokens, before passing them through the decoder blocks.

Let the teacher model produce hidden states from each decoder layer $l \in \{1, \ldots, L_t\}$, denoted as

$$[h_{l,1}^{\text{MLLM}}, \ldots, h_{l,N}^{\text{MLLM}}], \quad h_{l,n}^{\text{MLLM}} \in \mathbb{R}^{d_t}. \tag{4}$$

Note that $N$ is the sequence length, and $L_t$ and $d_t$ denote the number of decoder layers and the feature dimension of the teacher model, respectively.

**Token-wise Selection.** To identify a representative teacher token within the sequence, we use self-attention weights from each decoder layer. Let $\mathbf{A}_l^{(k)} \in \mathbb{R}^{N \times N}$ denote the attention matrix from head $k \in \{1, \ldots, K\}$ at layer $l$, where $K$ is the number of attention heads. We begin by averaging the attention weights across all heads:

$$\bar{\mathbf{A}}_l = \frac{1}{K} \sum_{k=1}^{K} \mathbf{A}_l^{(k)} \in \mathbb{R}^{N \times N}. \tag{5}$$

We then compute the maximum attention each token receives across all query positions:

$$s_{l,n} = \max_{i \in \{1, \ldots, N\}} \bar{\mathbf{A}}_l[i, n], \tag{6}$$

where $\bar{\mathbf{A}}_l[i, n]$ denotes the attention weight from token $i$ (as query) to token $n$ (as key). Finally, we select the token with the highest received attention:

$$n_l^* = \underset{n \in \{1,\ldots,N\}}{\arg\max} \; s_{l,n}, \quad h_l^{\text{Teach}} = h_{l,n_l^*}^{\text{MLLM}}. \tag{7}$$

This strategy selects the most attended token in each layer, under the assumption that such tokens are likely to carry salient multimodal information.

**Layer-wise Selection.** While token selection is guided by attention, we empirically select a subset of teacher layers for distillation to reduce computational cost. Let $\mathcal{S} = \{s_1, \ldots, s_{L_s}\} \subseteq \{1, \ldots, L_t\}$ denote the selected set of teacher layers, where $|S| = L_s$. We explore strategies such as selecting layers from fixed relative positions (e.g., sampling from early, middle, and late stages of the teacher) or using uniform stride intervals across the full depth. This design is motivated by the characteristic of Transformer representations, where lower layers tend to capture fine-grained information and higher layers encode abstract multimodal semantics. Based on empirical results, we adopt the stride-based selection strategy for all experiments.

### 3.3 LOSS FUNCTIONS

#### 3.3.1 CONTRASTIVE LOSS

Following the standard CLIP training (Radford et al., 2021), we use an InfoNCE loss to align image and text embeddings. Let $z_i^I$ and $z_i^T$ denote the image and text embeddings for the $i$-th sample in a batch of size $B$. The contrastive loss using image embeddings as anchors is given by:

$$\mathcal{L}_{\text{contrast}}^I = \frac{1}{B} \sum_{i=1}^{B} -\log \frac{\exp(\text{sim}(z_i^I, z_i^T)/\tau)}{\sum_{j=1}^{B} \exp(\text{sim}(z_i^I, z_j^T)/\tau)}, \tag{8}$$

where $\text{sim}(\cdot, \cdot)$ denotes cosine similarity and $\tau$ is a temperature. Similarly, we compute $\mathcal{L}_{\text{contrast}}^T$ by treating text embeddings as anchors. The final contrastive loss is given by:

$$\mathcal{L}_{\text{contrast}} = \frac{1}{2} \left( \mathcal{L}_{\text{contrast}}^I + \mathcal{L}_{\text{contrast}}^T \right). \tag{9}$$

#### 3.3.2 DISTILLATION LOSS

To align intermediate representations between the teacher and student models, we consider a set of similarity-based objectives for distillation. These objectives fall into two main categories:

- **Direct similarity loss:** such as mean squared error (MSE), which operates on a per-sample basis and compares feature vectors directly.
- **Relational similarity loss:** such as Centered Kernel Alignment (CKA) (Kornblith et al., 2019), which assesses structural similarity by aligning either the sample-level Gram matrices or the feature-level covariance matrices.

**Direct Similarity Loss.** Direct similarity losses enforce a sample-wise correspondence between the teacher's and the student's intermediate representations. Within a mini-batch size $B$, let $h_{l,i}^{\text{Stud},I}$ denote the student's image representation for the $i$-th sample from layer $l$, and $h_{s_l,i}^{\text{Teach},I}$ be the corresponding teacher's image representation from a selected teacher layer $s_l$. For each layer, we compute the similarity loss between the student and teacher representations for both image and text, and sum across layers. The resulting objective is averaged over the batch:

$$\mathcal{L}_{\text{distill}} = \frac{1}{B} \sum_{i=1}^{B} \frac{1}{L_s} \sum_{l=1}^{L_s} \frac{1}{2} \Big( \ell_{\text{direct}}(h_{l,i}^{\text{Stud},I}, h_{l,i}^{\text{Teach},I}) + \ell_{\text{direct}}(h_{l,i}^{\text{Stud},T}, h_{l,i}^{\text{Teach},T}) \Big), \tag{10}$$

where $L_s$ is the number of student layers. In our experiments, we consider the MSE, cosine similarity, KL divergence, and JS divergence loss as the direct loss functions, denoted as $\ell_{\text{direct}}$.

**Relational Similarity Loss.** We primarily adopt the CKA-based loss, which is effective for transferring knowledge between different architectures (Dasgupta & Cohn, 2025). Unlike direct similarity losses, CKA captures the global structural alignment of representations across a batch of samples. We first gather the layer-wise representations for all $B$ samples in a mini-batch into matrices. For a given modality, let $\mathbf{X}_l \in \mathbb{R}^{B \times D}$ be the matrix of student hidden states from layer $l$, and $\mathbf{Y}_{s_l} \in \mathbb{R}^{B \times D}$ be the matrix of corresponding teacher hidden states from layer $s_l$. CKA operates by comparing the Gram matrices of these centered feature matrices. First, the matrices are centered:

$$\tilde{\mathbf{X}}_l = \mathbf{X}_l - \frac{1}{B}\mathbf{1}\mathbf{1}^\top \mathbf{X}_l, \quad \tilde{\mathbf{Y}}_{s_l} = \mathbf{Y}_{s_l} - \frac{1}{B}\mathbf{1}\mathbf{1}^\top \mathbf{Y}_{s_l}, \tag{11}$$

where $\mathbf{1}$ is a column vector of ones. The Gram matrices, $K_l \in \mathbb{R}^{B \times B}$ and $L_{s_l} \in \mathbb{R}^{B \times B}$, are then computed:

$$\mathbf{K}_l = \tilde{\mathbf{X}}_l \tilde{\mathbf{X}}_l^\top, \quad \mathbf{L}_{s_l} = \tilde{\mathbf{Y}}_{s_l} \tilde{\mathbf{Y}}_{s_l}^\top. \tag{12}$$

Finally, CKA is calculated as the normalized Frobenius inner product of these Gram matrices:

$$\text{CKA}(\mathbf{X}_l, \mathbf{Y}_{s_l}) = \frac{\langle \mathbf{K}_l, \mathbf{L}_{s_l} \rangle_F}{\|\mathbf{K}_l\|_F \|\mathbf{L}_{s_l}\|_F}, \tag{13}$$

where $\| \cdot \|_F$ and $\langle \cdot, \cdot \rangle_F$ denote the Frobenius norm and Frobenius inner product, respectively. The CKA similarity score is converted into a loss for a single layer, $\ell_{\text{CKA}}$. We adopt a common variant that uses a square root, which can provide better gradient properties and create a more sensitive loss when the similarity is high. The total CKA loss is calculated by averaging the single-layer losses across both image and text modalities and summing them over a predefined set of layers:

$$\ell_{\text{CKA}}(\mathbf{X}_l, \mathbf{Y}_{s_l}) = 1 - \sqrt{\text{CKA}(\mathbf{X}_l, \mathbf{Y}_{s_l})}, \tag{14}$$

$$\mathcal{L}_{\text{distill}} = \frac{1}{L_s} \sum_{l=1}^{L_s} \frac{1}{2} \left( \ell_{\text{CKA}}(\mathbf{X}_l^I, \mathbf{Y}_{s_l}^I) + \ell_{\text{CKA}}(\mathbf{X}_l^T, \mathbf{Y}_{s_l}^T) \right). \tag{15}$$

The total training objective combines contrastive loss with the distillation loss:

$$\mathcal{L}_{\text{total}} = \mathcal{L}_{\text{contrast}} + \lambda \mathcal{L}_{\text{distill}}, \tag{16}$$

where $\lambda$ is a weighting factor.

## 4 EXPERIMENTS

### 4.1 EXPERIMENTAL SETTINGS

**Datasets.** We use CC3M (Sharma et al., 2018) as the pretraining dataset. We evaluate our approach on 11 compositionality benchmarks (Yuksekgonul et al., 2022; Ma et al., 2023; Hsieh et al., 2023; Parcalabescu et al., 2021; Zhao et al., 2022; Kamath et al., 2023; Krojer et al., 2022; Hendricks & Nematzadeh, 2021; Thrush et al., 2022; Wang et al., 2023; Peng et al., 2024), two image-text retrieval benchmarks (Chen et al., 2015; Plummer et al., 2015), and 13 classification datasets. Detailed descriptions of classification datasets are provided in Appendix B.1.

**Implementation Details.** As the student model, we employ a ViT-B/32 image encoder and a 12-layer Transformer text encoder, following the CLIP Base architecture. An auxiliary head is appended to each modality branch, consisting of a linear projection followed by Layer Normalization. For the teacher model, we use Qwen2.5-VL-3B, an MLLM with 36 Transformer decoder layers, selected based on the criteria described in Section 3.1. All models are trained with a global batch size of 4096 across 8 NVIDIA A100 GPUs. To ensure a fair comparison, any additional negative samples introduced by competing methods are included within the same overall batch size budget. Detailed hyperparameter configurations are provided in Appendix B.2.

Table 1: Performance on 11 compositionality benchmarks. Note that *SD* denotes the SDXL-Turbo model used to generate negative images.

| Method | External Source | ARO | CREPE | EQBEN | ImageCODE | SugarCrepe | SvoProbes | VALSE | VLChecklist | WhatsUp | Winoground | SPEC | Average |
|---|---|---|---|---|---|---|---|---|---|---|---|---|---|
| CLIP | – | 35.6 | 11.6 | 14.8 | 15.0 | 59.7 | 75.8 | 55.1 | 64.5 | 41.3 | 7.00 | 28.5 | 37.1 |
| LaCLIP | LLAMA-7B | 35.2 | 9.98 | 14.4 | 14.8 | 64.0 | 76.9 | 56.6 | 64.8 | 41.5 | 4.75 | 28.3 | 37.5 |
| NegCLIP | WordNet | 36.0 | 10.9 | 14.8 | 15.2 | 59.6 | 76.2 | 56.9 | 64.5 | 42.5 | 6.50 | 28.8 | 37.4 |
| NegCLIP | Qwen3-4B | 36.2 | 11.7 | 14.8 | 15.8 | 62.4 | 76.3 | 56.1 | 64.0 | 40.9 | **7.00** | 28.2 | 37.6 |
| FSC-CLIP | WordNet | 36.4 | 10.5 | 15.6 | 15.2 | 59.8 | 77.8 | 53.6 | 60.5 | 41.4 | 5.50 | 28.9 | 36.8 |
| FSC-CLIP | Qwen3-4B | 35.1 | 9.86 | 16.3 | 14.9 | 63.8 | 78.8 | 57.4 | 64.4 | 41.8 | 5.50 | **29.2** | 38.0 |
| TripletCLIP | Qwen3-4B, SD | 34.6 | 10.5 | 16.4 | 16.9 | 65.7 | 78.8 | 59.4 | 64.8 | 41.1 | 4.25 | 29.0 | 38.3 |
| MLLMCLIP | Qwen2.5-VL-3B | **36.6** | **11.8** | **17.0** | **17.2** | **67.5** | **78.9** | **59.5** | **65.8** | **42.8** | 6.50 | 29.0 | **39.3** |

Table 2: Zero-shot classification performance on 13 classification datasets.

| Method | External Source | Caltech 101 | CIFAR-10 | CIFAR-100 | DTD | EuroSAT | FER 2013 | Flower 102 | Food 101 | ImageNet | KITTI | Pet | RESISC45 | VOC 2007 | Average |
|---|---|---|---|---|---|---|---|---|---|---|---|---|---|---|---|
| CLIP | – | 37.8 | 53.3 | 21.7 | 7.93 | 12.1 | 12.4 | 8.88 | 8.91 | 12.4 | 20.0 | 9.95 | 15.9 | 54.0 | 21.2 |
| LaCLIP | LLAMA-7B | 41.7 | 49.7 | 21.2 | 11.2 | 14.3 | 19.5 | 11.2 | 9.34 | 13.7 | 30.7 | 17.1 | 17.1 | 60.9 | 23.9 |
| NegCLIP | WordNet | 41.5 | 52.1 | 24.4 | 8.62 | 19.0 | **22.7** | **11.5** | 9.12 | 13.8 | 21.9 | 9.36 | 21.2 | 58.7 | 24.1 |
| NegCLIP | Qwen3-4B | 41.4 | 55.3 | 24.1 | 10.6 | 18.3 | 18.9 | 9.16 | 9.49 | 13.7 | 30.4 | 8.81 | 17.9 | 56.8 | 24.2 |
| FSC-CLIP | WordNet | 42.3 | 58.3 | 24.2 | 9.15 | 23.6 | 17.4 | 10.4 | 8.18 | 13.6 | 25.6 | 9.54 | 21.2 | 59.5 | 24.8 |
| FSC-CLIP | Qwen3-4B | 41.5 | 54.2 | 25.1 | 9.84 | 18.0 | 16.8 | 9.03 | 9.40 | 13.5 | 20.8 | 10.1 | 20.3 | 63.1 | 24.0 |
| TripletCLIP | Qwen3-4B, SD | 42.4 | 46.5 | 22.3 | 13.5 | 24.6 | 17.2 | 10.4 | 9.97 | 14.2 | 23.9 | 10.8 | **22.8** | 56.3 | 24.2 |
| MLLMCLIP | Qwen2.5-VL-3B | **44.2** | **59.7** | **29.7** | **14.7** | **27.2** | 15.6 | 10.1 | **10.1** | **14.7** | **38.3** | 11.2 | 20.1 | **63.2** | **27.6** |

## 4.2 REPRODUCING PRIOR WORKS

To ensure a fair comparison, we re-implement several key baselines: CLIP (Radford et al., 2021), LaCLIP (Fan et al., 2023), NegCLIP (Yuksekgonul et al., 2022), FSC-CLIP (Oh et al., 2024), and TripletCLIP (Patel et al., 2024), which enhance CLIP training by incorporating additional supervision derived from external knowledge sources, such as WordNet (Fellbaum, 2010), LLM, and diffusion model. For LaCLIP, we use the publicly released LLaMA-generated positive captions[1]. For NegCLIP and FSC-CLIP, we reproduce WordNet-based hard negative caption construction as described in the respective papers. In addition, we extend both methods by generating alternative hard negatives using Qwen3-4B[2]. For TripletCLIP, we use the same Qwen3-4B-generated captions as prompts and feed them into the SDXL-Turbo model[3] to synthesize corresponding negative images. Note that we follow the prompt templates from TripletCLIP and generate one synthetic negative, either a caption or an image, for each corresponding sample in the CC3M dataset.

## 4.3 MAIN RESULTS

**Compositionality.** As shown in Table 1, all methods utilizing expert-driven supervision outperform the CLIP baseline, indicating the benefit of incorporating external knowledge. LaCLIP relies solely on additional positive texts and yields a slight improvement. Interestingly, while NegCLIP does not gain a significant advantage from using LLM-generated negatives over WordNet-based ones, FSC-CLIP achieves substantial improvements. TripletCLIP introduces synthetic negative images from a large-scale generative model, yet it provides limited benefit compared to approaches that rely solely on negative captions. Our method outperforms existing approaches on 9 out of 11 compositionality benchmarks, achieving the highest average score. The results demonstrate that compositional reasoning ability can be effectively strengthened by leveraging MLLM-derived signals rather than relying on explicit data generation.

---

[1] https://github.com/LijieFan/LaCLIP

[2] https://huggingface.co/Qwen/Qwen3-4B

[3] https://huggingface.co/stabilityai/sdxl-turbo

Table 3: Zero–shot retrieval performance on MSCOCO and Flickr-30K datasets.

| | | Image–to–text retrieval | | | | | | Text–to–image retrieval | | | | | |
| | | MSCOCO | | | Flickr-30K | | | MSCOCO | | | Flickr-30K | | |
| Method | External Source | R@1 | R@5 | R@10 | R@1 | R@5 | R@10 | R@1 | R@5 | R@10 | R@1 | R@5 | R@10 |
|---|---|---|---|---|---|---|---|---|---|---|---|---|---|
| CLIP | – | 9.02 | 24.3 | 33.7 | 18.3 | 39.9 | 50.4 | 6.85 | 18.7 | 27.1 | 12.9 | 31.6 | 41.5 |
| LaCLIP | LLAMA-7B | 9.80 | 24.8 | 33.7 | 20.8 | 40.4 | 51.5 | 6.72 | 19.1 | 27.8 | 15.9 | 36.8 | 47.8 |
| NegCLIP | WordNet | 11.7 | 27.9 | 38.7 | 23.0 | 48.4 | 58.0 | 8.24 | 21.9 | 31.1 | 16.2 | 36.6 | 47.5 |
| NegCLIP | Qwen3-4B | 9.38 | 24.4 | 34.4 | 20.1 | 42.8 | 53.6 | 7.66 | 20.0 | 28.5 | 13.7 | 32.2 | 42.7 |
| FSCCLIP | WordNet | 11.8 | 28.1 | 39.7 | 22.7 | 46.2 | 58.2 | 8.42 | 22.7 | 31.9 | 16.1 | 36.8 | 48.4 |
| FSCCLIP | Qwen3-4B | 11.4 | 27.8 | 37.7 | 22.0 | 46.7 | 58.0 | 8.91 | 23.5 | 32.8 | 17.4 | 38.4 | 49.0 |
| TripletCLIP | Qwen3-4B, SD | 12.3 | 29.9 | 40.3 | 22.9 | 46.8 | 59.6 | 9.11 | 23.5 | 33.1 | 17.8 | 39.1 | 49.8 |
| MLLMCLIP | Qwen2.5-VL-3B | **12.6** | **30.0** | **40.7** | **26.5** | **51.7** | **61.3** | **9.37** | **24.7** | **34.4** | **18.9** | **41.7** | **52.8** |

Table 4: Effect of token selection strategies on downstream tasks. The first two columns indicate the positions of teacher tokens used for distillation. `[Image End]` and `[Text End]` refer to the token positions at the end of the image segment and the full multimodal sequence, respectively.

| **Image Teacher** | **Text Teacher** | **Comp.** | **Zero-shot Cls.** | **I2T Ret.** | **T2I Ret.** |
|---|---|---|---|---|---|
| None | None | 37.1 | 21.2 | 13.7 | 9.88 |
| `Image End` | None | 37.7 | 23.6 | 15.6 | 11.8 |
| `Text End` | None | 38.2 | 24.3 | 16.6 | 12.3 |
| `Text End` | `Text End` | 38.5 | 25.6 | 16.9 | 12.6 |
| Attention-based Selection | | **39.3** | **27.6** | **19.6** | **14.1** |

**Zero-shot Classification & Retrieval.** Tables 2 and 3 report the zero-shot classification and retrieval performance, respectively. NegCLIP and FSC-CLIP do not exhibit clear gains when WordNet-based negatives are replaced with captions generated by recent LLMs (Qwen3-4B), suggesting a potential limitation in these frameworks to harness the richer knowledge of advanced language models. Similarly, TripletCLIP shows no clear performance advantage despite the additional computational overhead of its image generation step. In contrast, MLLMCLIP achieves state-of-the-art performance on 10 out of 13 datasets with a substantially higher average accuracy. These results confirm that MLLM-guided distillation not only improves compositionality but also provides clear advantages in generalizability.

## 4.4 ABLATION STUDIES

We report ablation results across four evaluation groups. Compositionality is measured as the average accuracy over 11 datasets and denoted as **Comp.**. For zero-shot classification, we report the average performance over 13 datasets as **Zero-shot Cls.**. For retrieval, we report the average Recall@1 on two datasets for Image-to-Text (**I2T**) and Text-to-Image (**T2I**), respectively.

**Teacher Token Selection.** Table 4 reports the effect of different teacher token selection strategies. We first evaluate several fixed-position baselines using tokens from the end of the image segment (`[Image End]`) or the multimodal sequence (`[Text End]`). Due to the causal masking strategy of MLLM, the `[Image End]` token encapsulates unimodal visual understanding, while the `[Text End]` token captures multimodal understanding. Using the `[Image End]` token as the image teacher already improves performance over the baseline, indicating the strong unimodal capabilities of the MLLM. Performance further improves when we replace the image teacher with the `[Text End]` token, as multimodal information is incorporated into the guidance. Adding the `[Text End]` token as an additional text teacher yields another performance gain, confirming that supervising both modalities is beneficial. Finally, the attention-based selection strategy outperforms all fixed-position baselines across all metrics. This demonstrates the benefit of adaptively choosing informative tokens from each layer and confirms that relying on a fixed token is insufficient to capture the rich semantics embedded in the MLLM's representations.

Table 5: Effect of different distillation loss functions on downstream performance.

| Loss Type | Loss Function | Comp. | Zero-shot Cls. | I2T Ret. | T2I Ret. |
|---|---|---|---|---|---|
| Direct | MSE | 30.7 | 7.00 | 0.06 | 0.06 |
| | Cosine | 38.6 | 25.9 | 18.3 | 13.4 |
| | KL Divergence | 37.8 | 26.0 | 17.6 | 13.0 |
| | JS Divergence | 38.1 | 23.8 | 16.5 | 12.0 |
| Relational | CKA | **39.3** | **27.6** | **19.6** | **14.1** |

Table 6: Downstream performance with varying teacher layer selection strategies for distillation.

| Layer Selection | Comp. | Zero-shot Cls. | I2T Ret. | T2I Ret. |
|---|---|---|---|---|
| Lower Block (1–12) | 38.5 | 26.8 | 17.5 | 12.9 |
| Middle Block (13–24) | 38.2 | 26.2 | 17.4 | 12.9 |
| Upper Block (25–36) | 38.4 | 26.5 | 17.9 | 12.6 |
| Strided (1,4,7...) | **39.3** | **27.6** | **19.6** | **14.1** |

**Loss Functions.** Table 5 presents the effect of different distillation loss functions. Mean Squared Error (MSE) enforces a strict element-wise match between feature vectors, resulting in performance collapse. This stems from the fundamental architectural mismatch between the generative decoder and the representational encoder, which produces feature spaces with incompatible scales. Scale-normalized objectives alleviate this issue and stabilize training, but remain suboptimal since they focus only on pointwise alignment. Notably, CKA consistently outperforms direct similarity approaches across all downstream tasks. This result strongly suggests that preserving the structural relationships among samples is a more robust and effective method for knowledge transfer than per-sample feature matching, especially in a cross-architecture setting.

**Teacher Layer Selection.** We investigate the optimal strategy for selecting teacher layers for distillation, with results presented in Table 6. We first evaluate strategies that use contiguous blocks of layers from the teacher MLLM: the lower (1–12), middle (13–24), and upper (25–36). These block-based strategies yield comparable performance, with no clear advantage for any single block. In contrast, a strided selection strategy that samples layers uniformly across the entire network (e.g., 1, 4, 7, ...) outperforms all block-based approaches. This indicates that supervision derived from a broader range of layers provides more comprehensive guidance, whereas relying on a contiguous block of layers may overlook information distributed throughout the teacher model.

## 5 CONCLUSION

In this work, we investigate effective strategies for distilling multimodal interaction signals from MLLMs into CLIP. Our framework bridges the fundamental architectural mismatch between these models through three key contributions: an MLLM evaluation protocol, attention-based token selection, and a structure-aware distillation loss. Our approach achieves strong performance across compositionality, zero-shot classification, and retrieval tasks, demonstrating the potential of MLLM-guided distillation for building compact and transferable vision-language encoders. Furthermore, our framework is model-agnostic and can be applied to a wide range of decoder-based MLLM and encoder-based vision-language models.

## LLM USAGE

During the preparation of this paper, we use large language models in a limited and assistive manner. For implementation, an LLM is utilized for code review and the detection of minor bugs. For writing, an LLM is used for English proofreading and grammar checks. We do not have LLMs draft whole passages or sentences from scratch, nor do we rely on them to generate novel methods or results.

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

Table A: Comparison of MLLM performance on the SugarCrepe benchmark.

| Model | GT-position | Replace | | | Swap | | Add | | Average |
|---|---|---|---|---|---|---|---|---|---|
| | | Object | Attribute | Relation | Object | Attribute | Object | Attribute | |
| LLAVA-1.6-mistral-7B | First | 99.8 | 99.6 | 98.9 | 99.2 | 99.8 | 98.5 | 98.6 | 99.2 |
| | Second | 81.8 | 57.2 | 50.8 | 20.8 | 23.7 | 59.4 | 29.6 | 46.2 |
| | Average | 90.8 | 78.4 | 74.9 | 60.0 | 61.8 | 78.9 | 64.1 | 72.7 |
| LLAVA-1.6-vicuna-7B | First | 99.4 | 99.8 | 96.9 | 95.9 | 98.8 | 97.0 | 95.1 | 97.6 |
| | Second | 82.5 | 55.3 | 56.1 | 30.2 | 27.9 | 40.8 | 21.7 | 44.9 |
| | Average | 91.0 | 77.5 | 76.5 | 63.1 | 63.4 | 68.9 | 58.4 | 71.2 |
| Qwen2-VL-2B | First | 96.5 | 90.6 | 85.9 | 80.4 | 91.4 | 94.2 | 81.9 | 88.7 |
| | Second | 98.2 | 95.6 | 92.3 | 90.2 | 92.2 | 96.5 | 93.5 | 94.1 |
| | Average | 97.3 | 93.1 | 89.1 | 85.3 | 91.8 | 95.4 | 87.7 | 91.4 |
| Qwen2-VL-7B | First | 99.1 | 95.8 | 94.0 | 93.1 | 96.9 | 95.5 | 87.1 | 94.5 |
| | Second | 98.7 | 96.5 | 94.2 | 92.7 | 97.9 | 97.2 | 94.5 | 96.0 |
| | Average | 98.9 | 96.1 | 94.1 | 92.9 | 97.4 | 96.3 | 90.8 | 95.2 |
| Qwen2.5-VL-3B | First | 99.0 | 95.6 | 92.0 | 93.9 | 97.6 | 96.2 | 90.9 | 95.0 |
| | Second | 98.2 | 94.0 | 88.8 | 91.0 | 96.0 | 97.2 | 92.8 | 94.0 |
| | Average | 98.6 | 94.8 | 90.4 | 92.5 | 96.8 | 96.7 | 91.8 | 94.5 |
| Qwen2.5-VL-7B | First | 99.1 | 98.5 | 96.2 | 95.1 | 99.3 | 98.2 | 96.7 | 97.5 |
| | Second | 97.9 | 92.4 | 88.2 | 87.4 | 94.1 | 94.0 | 79.1 | 90.4 |
| | Average | 98.5 | 95.4 | 92.2 | 90.8 | 96.7 | 96.2 | 87.9 | 94.0 |
| InternVL3-1B | First | 92.1 | 81.7 | 67.1 | 42.5 | 64.9 | 86.6 | 72.1 | 72.4 |
| | Second | 97.0 | 88.6 | 90.0 | 85.3 | 88.0 | 96.9 | 92.1 | 91.1 |
| | Average | 94.6 | 85.2 | 78.6 | 63.9 | 76.4 | 91.7 | 82.1 | 81.8 |
| InternVL3-2B | First | 97.5 | 95.7 | 91.5 | 86.9 | 95.2 | 95.5 | 87.1 | 92.8 |
| | Second | 97.3 | 94.0 | 88.5 | 83.3 | 88.4 | 93.8 | 86.9 | 90.3 |
| | Average | 97.4 | 94.9 | 90.0 | 85.1 | 91.8 | 94.6 | 87.0 | 91.6 |
| InternVL3-8B | First | 99.0 | 96.2 | 93.9 | 96.3 | 98.2 | 97.0 | 89.9 | 95.8 |
| | Second | 98.2 | 96.2 | 91.4 | 90.2 | 96.9 | 95.7 | 90.3 | 94.1 |
| | Average | 98.6 | 96.2 | 92.6 | 93.3 | 97.5 | 96.4 | 90.1 | 95.0 |
| LLAMA-3.2-Vision-11B | First | 87.0 | 85.1 | 77.3 | 84.9 | 88.0 | 82.5 | 74.7 | 82.8 |
| | Second | 85.5 | 77.7 | 77.2 | 65.3 | 81.5 | 87.1 | 76.3 | 78.7 |
| | Average | 86.2 | 81.4 | 77.3 | 75.1 | 84.8 | 84.8 | 75.5 | 80.7 |

Table B: Downstream performance with varying weight parameter during pre-training.

| $\lambda$ | Comp. | Zero-shot Cls. | I2T Ret. | T2I Ret. |
|---|---|---|---|---|
| 0.1 | 37.7 | 23.7 | 14.0 | 10.9 |
| 1 | **39.3** | **27.6** | **19.6** | **14.1** |
| 10 | 38.1 | 25.8 | 16.2 | 12.3 |
| 100 | 37.8 | 23.5 | 18.1 | 12.8 |

## A  ADDITIONAL ABLATION STUDIES

**Weight Parameter.**  Table B shows the impact of varying the weight $\lambda$ applied to the distillation loss. We observe that $\lambda = 1$ achieves the best overall performance. A smaller weight ($\lambda = 0.1$) provides an insufficient supervisory signal from the teacher, leading to degraded performance. Conversely, larger weights ($\lambda = 10$ and $100$) also harm performance, suggesting that an overly strong distillation objective may interfere with the model's primary contrastive learning task.

## B  EXPERIMENTAL SETTINGS

### B.1  CLASSIFICATION DATASETS

We evaluate zero-shot classification on 13 datasets, including: Caltech101 (Fei-Fei et al., 2007); CIFAR-10, CIFAR-100 (Krizhevsky, 2009); Describable Textures (Cimpoi et al., 2014); EuroSAT-CLIP Helber et al. (2019); FER-2013 (Goodfellow et al., 2013); Flower102 (Nilsback & Zisserman, 2008); Food101 (Bossard et al., 2014); ImageNet-1K (Deng et al., 2009); KITTI-Distance (Geiger

et al., 2013); Oxford-IIIT Pet (Parkhi et al., 2012); RESISC45-CLIP (Cheng et al., 2017); and PASCAL VOC2007 (Everingham et al., 2010).

## B.2 IMPLEMENTATION DETAILS

**Training Hyperparameters.** We train all models using the AdamW optimizer with a batch size of 4096, an initial learning rate of $5 \times 10^{-4}$, and a weight decay of 0.5 for 30 epochs. A cosine learning rate scheduler is applied with a linear warmup during the first epoch, where the learning rate increases from $1 \times 10^{-6}$ to the base learning rate and decays to $1 \times 10^{-5}$ by the end of training. All experiments are conducted using `bfloat16` precision. Unless otherwise specified, we use a fixed loss weight of $\lambda = 1$ for balancing the distillation and contrastive losses.

**Text Prompt Template.** For the MLLM teacher, we format the text input using a structured template designed to encourage compositional reasoning. This template is applied to the caption side of each image-caption pair used in contrastive learning. The full template is shown below:

```
Given the image and the caption {caption}, analyze whether this
caption accurately describes the image.  If it does, imagine a
similar caption that could be easily confused with it but is subtly
incorrect or misleading.  Internally reason about the difference
between the correct and incorrect caption, highlighting the key
visual-semantic concepts that make the original caption more
accurate.  Focus on compositional elements such as object attributes,
actions, relationships, and spatial arrangements.  Use this reasoning
to build an internal representation of the image that emphasizes
these distinctions.
```

## LIMITATION

While our work demonstrates the significant potential of MLLM to CLIP distillation, it has a few limitations that present opportunities for future research. The capabilities of the teacher MLLM fundamentally cap the performance of MLLMCLIP. Any biases, factual inaccuracies, or reasoning failures inherent in the teacher model can be transferred to the student during distillation. Although we propose a rigorous protocol for teacher selection, the optimal teacher may vary depending on the specific downstream tasks.

## REPRODUCIBILITY STATEMENT

We provide all implementation details, including model architectures, training hyperparameters, and evaluation protocols, in the main paper and appendix. All experiments are conducted with publicly available datasets, and we will release our code, pretrained models, and data processing scripts to ensure reproducibility.

