# OpenReview forum: "MLLMCLIP: Feature-Level Distillation of MLLM for Robust Vision-Language Representations"
_ICLR.cc/2026/Conference — ICLR 2026 Conference Withdrawn Submission_

### Official Review · Reviewer_Xxw5 · 2025-10-29

**Soundness:** 2
**Presentation:** 2
**Contribution:** 2
**Rating:** 2
**Confidence:** 5

**Summary:**

This paper distills an MLLM’s compositional reasoning ability into a CLIP encoder by selecting the most attended decoder token from each teacher layer and aligning student features with CKA, avoiding synthetic negatives while preserving dual-encoder efficiency.

**Strengths:**

1. Clear problem framing and practical motivation: improve compositionality without a synthetic data pipeline, and keep an efficient encoder at inference.
2. This method achieves SOTA on multiple compositionality benchmarks and consistent gains in zero-shot classification and retrieval; ablations highlight attention-based token selection and CKA as key to stable cross-architecture distillation.

**Weaknesses:**

1. The method is a careful assembly of known parts (KD + attention heuristics + CKA), rather than introducing a fundamentally new learning objective or architecture. The combination feels unsurprising with limited insigh.
2. The attention-based “pick one token per layer” strategy is plausible but quite heuristic. No qualitative analysis shows which tokens are chosen or whether they align with compositional elements.
2. Qwen2.5-VL-3B is chosen for performance/efficiency, but do not provide a systematic “teacher quality vs student skill” study (e.g., 3B vs 7B vs 8B). Would better teachers yield proportionally better students?
3. Claiming to “bypass generation pipeline inefficiency” is fair, but per-sample teacher forward passes during training are still expensive. A training-time compute comparison vs synthetic-negative pipelines would clarify the practicality.

**Questions:**

1, Why restrict to a single token per layer? Have you tried:
Top-k tokens by attention, attention-weighted pooling, or alternative saliency signals (e.g., gradient times input)?
Selecting separate tokens for image-specific vs text-specific supervision at each layer?
2. How sensitive is CKA distillation to batch size and distribution? Can you stabilize CKA for smaller batches via cross-batch memory or running-estimate Gram matrices?
3. How robust are your results across different student backbones (e.g., RN50, ViT-B/16, ViT-L/14) and larger pretraining corpora?

---

### Official Review · Reviewer_LKhn · 2025-10-31

**Soundness:** 1
**Presentation:** 2
**Contribution:** 2
**Rating:** 2
**Confidence:** 4

**Summary:**

This paper proposes MLLMCLIP, a CLIP model that distills knowledge from a much larger MLLM (Qwen2.5-VL-3B). Qwen2.5-VL-3B is selected among other multimodal architectures as it performs the best in a modified version of the SugarCrepe benchmark. Direct and relational similarity losses are explored for distillation.

**Strengths:**

1. The paper shows that some of the features in MLLMs are beneficial for simpler models as CLIP.

**Weaknesses:**

Minor Flaws

1. Use Self-contained captions: captions for figures and tables should provide an outline of the most relevant information in the figure or the table.  Figure 2 and tables 1, 2, 3, 5 and 6  provide little information.
2. Figure 2 is hard to parse and several of its elements are not explained in the caption and the text. For example what exactly is the color code, and what is the overall flow of information in this figure. Aux^{t} and Aux^{I} seem to be relevant elements in MLLMCLIPm, but they are not shown in the figure. The same happens with z^{I} and z^{T}
3. Line 187 what final-layer representations do the authors refer to? CLIP layers? MLLM layers? a specific  component layer?
4. Equation 2 and line 192 define R^{D} and R^{d} . I can't tell if these are two different dimensions or this is just a typo.
5. Revisit the notation in line 260 and 261, a variable name with 2 super index and two subindexes is overly complicated and hard to recall
6. The assertion in 221 must be toned down, as outlined later in section 4.4 not every token contains multi-modal information.

Related work

7. Related work should be re-writtent to directly address the most related world (i.e. those works who augment CLIP by means of distillation) and to briefly discuss the differences or advantages of MLLMCLIP.
8. The assertion “Our framework is agnostic to these architectural choices, capable of leveraging the multimodal understanding from both encoder-based and encoder-free MLLMs as a teacher.” is unsubstantiated, MLLMCLIP was only validated with Qwen2.5-VL-3B, and there is no empirical evidence that it can work across multiple designs of MLLMs.

Teacher model selection

9. Could the authors provide additional details on the experimental setup for the model selection, for example total number of questions, random baselines. Section 3,1 seems to imply that the entire SugarCrepe benchmark was used for this process.

10. Finally, the Qwen2.5-VL-3B is selected as it has the best performance in the SugarCrepe benchmark. It is expected that a distillation of this model should also have high performance on it. For fairness the SugarCrepe benchmark should be removed from the empirical evaluation.

Flawed Empirical evaluation:

11. This paper does not provide a direct and fair comparison against the state of the art. While LaCLIP, NegCLIP, FSC-CLIP and TripletCLIP retrain or fine-tune from augmented data, MLLMCLIP distills the feature representation from a much larger MLLM. These two tasks are fundamentally different.

12. At training time MLLMCLIP benefits from the learned feature representation of a 3 Billion model that's been trained over 4.1 trillions of language tokens, and a large set of multi-modal data comprising captions and instructional data, a significant portion of this data undergoes several stages of curation and clean-up. The selected baselines fine-tune over a fraction of the original CLIP training data without the presence of another model and most of the curation and extension is done automatically. When compared to MLLMCLIP, every baseline is at a significant disadvantage regarding the number of training data points, the size of the architectures involved at training time, and the level of data curation.
Given the significant mismatch in the training of the models, I don't think any of the results in tables 1,2 and 3 offers a fair comparison.
If the authors focus on distillation, there are far more suitable baselines, for example, [A], [B] and [C] already propose to distill larger CLIP models and text to image models.

[A] Yang, Chuanguang, et al. "Clip-kd: An empirical study of clip model distillation." Proceedings of the IEEE/CVF Conference on Computer Vision and Pattern Recognition. 2024

[B] Distilling Knowledge from Text-to-Image Generative Models Improves Visio-Linguistic Reasoning in CLIP (Basu et al., EMNLP 2024)

[C]Chen, Yifan, et al. "Comkd-clip: Comprehensive knowledge distillation for contrastive language-image pre-traning model." arXiv preprint arXiv:2408.04145 (2024).

Potential Data Leakage

13. The training set of Qwen2.5-VL-3B is not publicly available, it is possible that the validation data contained in the selected benchmarks was previously seen by Qwen2.5-VL-3B at training time (specially datasets with captioned images as COCO and Flickr-30K). As the authors resort to feature distillation, MLLMCLIP has potentially distilled features obtained from the datasets used in tables 1, 2 and 3. This is yet another unfair advantage for MLLMCLIP.

Limited Novelty

14. Finally, I don't see a clear contribution or innovation from this paper. None of the selected distillation strategies is novel, and the model selection seems to be a rather simple adaptation of the SugarCrepe benchmark. The improved results in the benchmarks are better attributed to the large scale of the distilled model and its training data, the unfair baseline selection, and the possible data leakage on the evaluation benchmarks.

**Questions:**

1. I would like to know how the authors can find corresponding language tokens between the LLM in Qwen2.5-VL-3B and the language model in CLIP. My understanding is that Qwen2.5-VL-3B has about 151k unique tokens while CLIP has about 50K. I’ll be surprised if an individual sentence has the exact same amount of tokens in both models.
2. In a similar spirit does the patch size in the visual embedding of Qwen2.5-VL-3B and the patch size in CLIP visual embedding match? Do the authors perform some upsampling to obtain matching visual patches?
3. Line 353, Could the authors explain why all the baselines (including CLIP) must be re-implemented? What design choice had to be aligned to better compare with  MLLMCLIP?

---

### Official Review · Reviewer_gDAq · 2025-11-01

**Soundness:** 2
**Presentation:** 2
**Contribution:** 2
**Rating:** 4
**Confidence:** 4

**Summary:**

The paper introduces a student-teacher distillation framework, MLLMCLIP, where CLIP is considered a student and features are distilled from an MLLM to improve CLIP's performance on compositionality and zero-shot retrieval tasks. The paper shows experiments on different benchmarks and provides a variety of ablations to support their claims.

**Strengths:**

1. The paper is well-written and easy to understand.
2. The authors provide results over a variety of compositionality benchmarks and perform ablation to strengthen the design choices in MLLMCLIP.
3. The idea to directly use MLLM features as knowledge from the teacher model is novel.

**Weaknesses:**

1. It’s a bit unclear why performing distillation from synthetic data generated using an MLLM is inefficient. Please provide more results or an explanation to support this claim.
2. For token selection, it’s a bit unclear why only the maximum attention token would be suitable as a teacher token. Can you show what happens when you increase the number of teacher tokens by also incorporating lesser attention tokens?
3. It's also unclear if the performance of the student model really depends on how the teacher model performs on the multimodal prompt? Please provide results for MLLMCLIP when other teacher models is used
4. No qualitative examples comparing MLLMCLIP with other methods.
5. Are the “Image End” and “Text End” taken from assumed to be part of MLLM’s vocabulary or added separately? If added separately, does the MLLM require some fine-tuning for these new tokens?
6. For MLLMs like LLaVA-v1.5, where the vision encoder is already a CLIP model, would such an MLLM be of any benefit as a teacher for the same CLIP-based student? If not, please mention what kind of MLLMs the MLLMCLIP distillation would be helpful for?
7. Does performing distillation with MLLMCLIP require more computation than other synthetic data generation-based works?

**Questions:**

Please refer to the weaknesses for the questions.

---

### Official Review · Reviewer_aah4 · 2025-11-02

**Soundness:** 2
**Presentation:** 2
**Contribution:** 3
**Rating:** 4
**Confidence:** 4

**Summary:**

The paper proposes MLLMCLIP, a feature-level distillation framework that enhances vision–language models like CLIP by directly transferring multimodal knowledge from a Multimodal Large Language Model (MLLM). The method maps representations of different layers of the MLLM to layer representation of the CLIP models and optimizes their alignment with a Centered Kernel Alignment for knowledge transfer. MLLMCLIP achieves good results on 9 out of 11 compositionality benchmarks and improving general zero-shot tasks.

**Strengths:**

- Direct distillation of representations from MLLMs into more efficient CLIP models is an important research direction as it circumvents expensive synthetic data generation. This is a generally underexplored direction.
- The chosen architecture and loss function are reasonable. The CKA loss is shown to provide benefits over other options, such as MSE, Cosine, KL/JS divergence.
- The results demonstrate an improvement over alternative approaches for equal training and architecture settings.

**Weaknesses:**

- There is missing literature that is relevant to this line of research.
  * There are CLIP variants that focus on more fine-grained understanding such as DreamLIP [A], or FLAIR [B]. Both use synthetic data which could serve as a comparison to this alternative direction.
  * There is an increasing amount of work that tries to turn MLLMs into embedding models, such as VLM2Vec [C], or LamRA [D]. It would be beneficial to position the paper in contrast to this approach as both directions try to achieve the same goal.

- Some method design choices could be better justified.
  * It is not obvious why any of the tested layer selection strategies would be optimal for distillation. It was not tested if all CLIP layers require alignment, or if the same MLLM layer could be chosen instead of a set of sequential/strided layers. One sensible alternative would be to simple distill one MLLM layer representation to one (last or penultimate) layer of CLIP.
  * The texts describes the teacher embedding for the image $h^{\text{Teach},I}$ separately from the text one $h^{\text{Teach},T}$. It is not explained how they are chosen in practice. Are they simply identical? If so, why make this distinction?

- The absolute performance of the presented results is quite poor.
  * All presented results on the different tasks lack behind severely the state-of-the-art. For example for Flickr-30K R@1 retrieval, MLLMCLIP reports 18.9 (T2I) and 26.5 (I2T), which is significantly lower than the results from FLAIR (81.1, 94.7) and VLM2Vec (88.1, 97.6). It is also not clear why the base CLIP performance is so low when OpenCLIP much better performance (71.9, 87.5, taken from [B]).
  * It is not clear whether the paper reproduces all the results of the baselines or if existing checkpoints were used.
  * There is no reasoning given for the chosen model size of ViT-B/32. Typically ViT-B/16 is much more competitive for the same number of parameters. In general, the paper could have much higher impact if it tried to get closer to SOTA results, although I can understand that resources can be a limiting factor.

- The teacher token selection ablation (Tab. 4) raises questions.
  * How is the teacher and text token chosen for "Attention-based Selection"? (related to earlier remark on missing clarity)
  * When both image and text teacher is "None", does it mean that no distillation is performed and only the CLIP loss is used? In this case, the performance improvement, while present, is rather small for the compositionality benchmarks, although this was a core motivation for this approach.

Minor:
- Fig. 1 and Sec. 3.1 would better fit into the start of the experiments section as they have little to do with the method. This will become increasingly irrelevant as MLLMs become stronger.

[A] Zheng et al., DreamLIP: Language-Image Pre-training with Long Captions, ECCV 2024
[B] Xiao et al., FLAIR: VLM with Fine-grained Language-informed Image Representations, CVPR 2025
[C] Jiang et al., VLM2Vec: Training Vision-Language Models for Massive Multimodal Embedding Tasks, ICLR 2025
[D] Liu et al., LamRA: Large Multimodal Model as Your Advanced Retrieval Assistant, CVPR 2025

**Questions:**

Please refer to the weaknesses section. Questions of particular interest are:
- How do you position MLLMCLIP with respect to [A,B,C,D]?
- What is the reasoning behind having a distillation loss on all CLIP layers? Is there any empirical evidence that suggests this is the best option?
- How do you explain the performance discrepancy between your presented results and results from the literature?
- Did you reproduce the baselines or use model checkpoints?
- Why was ViT-B/32 chosen? What prevents you from scaling to more competitive sizes/architectures (e.g. ViT-B/16)?
- How do you explain that the core distillation contribution does not improve compositionality results as much as classification and retrieval?

---

### Note · Authors · 2025-11-24

I have read and agree with the venue's withdrawal policy on behalf of myself and my co-authors.